# Metal Embedded Phthalocyanine Monolayers as Promising Materials for Toxic Formaldehyde Gas Detection: Insights from DFT Calculations

**Rou Xue** [1], **Chen Wang** [1,2], **Yajun Wang** [1,2], **Qijun Guo** [1], **Enrui Dai** [1] and **Zhifeng Nie** [1,*]

[1] Yunnan Key Laboratory of Metal-Organic Molecular Materials and Device, School of Chemistry and Chemical Engineering, Kunming University, Kunming 650214, China
[2] School of Physical Science and Technology, Kunming University, Kunming 650214, China
* Correspondence: niezf123@163.com

**Abstract:** The design of the good-performance materials for toxic formaldehyde ($CH_2O$)-gas-detection is critical for environmental preservation and human health. In this work, density functional theory (DFT) calculations were employed to investigate the adsorption behavior and electronic properties of $CH_2O$ on transition metal (TM)-doped phthalocyanine monolayers. Our results prove that PdPc and RuPc monolayers are thermodynamically stable. Analysis of the adsorption energy showed that the $CH_2O$ gas molecule was chemisorbed on the RuPc monolayer, while it was physisorbed on the PdPc nanosheet. The microcosmic interaction mechanism within the gas-adsorbent system was revealed by analyzing the density of states, the charge-density difference, the electron-density distribution, and the Hirshfeld charge transfer. Additionally, the RuPc monolayer was highly sensitive to $CH_2O$ due to the obvious changes in electrical conductivity, and the recovery time of $CH_2O$ molecule was predicted to be 2427 s at room temperature. Therefore, the RuPc monolayer can be regarded as a promising gas-sensing material for $CH_2O$ detection.

**Keywords:** transition metal phthalocyanine; formaldehyde; DFT computations; surface adsorption

## 1. Introduction

Formaldehyde ($CH_2O$), a common a chemical compound, is applied widely in the field of industrial applications, such as furniture, wood processing, textiles, and chemical production [1,2]. However, in the environment, $CH_2O$ is also an extremely poisonous, colorless, odorless highly volatile gas, which can cause health problems such as allergy, poisoning, pulmonary damage, and cancer [3–5]. Short-term exposure leads to respiratory-tract infections and skin allergies even at low concentration levels. Therefore, from environmental and health points of view [6–9], monitoring and controlling the emissions of $CH_2O$ is essential and can be achieved through developing an advanced gas sensor with good performance.

Over the past few decades, several metal oxide semiconductors (MOS), such as $SnO_2$ [10,11], $WO_3$ [12], $In_2O_3$ [13,14], $ZnO$ [15,16], and $TiO_2$ [17,18] have been proved effective in formaldehyde detection. However, some drawbacks such as poor sensitivity, weak adsorption ability, high cost, and long response speed, restrict their industrial application. Recently, two-dimensional phthalocyanine (Pc) materials have attracted widespread attention from researchers due to their unique properties [19–21]. Interestingly, phthalocyanine exhibits diverse morphologies and properties such as high specific surface area, fascinating charge transfer properties, and excellent optical performance. These novel behaviors have made it possible to use phthalocyanine for advanced technological applications such as photovoltaics [22], optoelectronics [23], electrocatalysis [24], and spintronics [25]. Resistive phthalocyanine-materials-based gas sensors have been successfully developed in the experiments [26–28], however, their poor stability/sensitivity/selectivity restricts their

further industrial application. To overcome these limitations, the Pc is modified by introducing transition metal atoms on the surface which can significantly improve the sensing properties [29–31].

Recently, many valuable contributions have been made by studies conducted into the doping of metal atoms on phthalocyanine for sensing common gas molecules such as $CO_2$, CO, NO, $O_3$, and $O_2$. Dopant atoms (e.g., Sc, Ti, Fe, and Zn) can significantly alter the electronic and magnetic properties of Pc and, correspondingly, could promote its adsorption performance. For instance, Lü et al. [32] employed DFT calculations and Monte Carlo simulations to investigate the $CO_2$ adsorption on TMPc (TM = Sc, Ti, and Fe) frameworks, and showed that the ScPc sheet exhibits excellent selectivity towards $CO_2$. Using the DFT method, zinc phthalocyanine (ZnPc) and its derivatives were also illustrated to be highly sensitive in the detection of $CO_2$, CO, NO, $O_3$, and $O_2$ gas molecules [33]. However, the sensing performance of metal-embedded phthalocyanine monolayers on $CH_2O$ gas has rarely been reported.

In this work, we performed the DFT calculation to determine the gas-sensing performance of palladium phthalocyanine (PdPc) and ruthenium phthalocyanine (RuPc) monolayers towards the $CH_2O$ gas molecule. The stable adsorption configurations, adsorption behaviors, and electronic characteristics of the $CH_2O$ adsorbed on the Pc, PdPc, and RuPc nanosheets were systematically investigated. The density of states (DOSs), charge-density difference (CDD), electron-density distribution (EDD), and charge transfer of different gas-adsorbent systems were further analyzed. The adsorption ability, sensing performance, and recovery time were evaluated to explore the possibility of PdPc and RuPc monolayers as promising gas-sensing materials for $CH_2O$ gas detection.

## 2. Modeling and Computing

The adsorption characteristics of corresponding adsorption systems were calculated by the $Dmol^3$ module of the Materials Studio 8.0 Package based on the density functional theory (DFT) [34]. The Perdew–Burke–Ernzerhof (PBE) function within the generalized gradient approximation was used to describe the exchange-correlation effect of electrons [35]. The weak long-range interactions were corrected using the dispersion-corrected DFT (DFT-D) approach proposed by Grimme [36]. During the simulation, we adopted the double numerical plus polarization basis set and employed the DFT semi-core pseudopotential (DSSP) method [37,38]. A Monkhorst–Pack k-point mesh of $6 \times 6 \times 1$ ($12 \times 12 \times 1$) was chosen for structural optimization (electronic properties calculation) [39]. The corresponding energy tolerance accuracy, maximum force, and displacement were selected as $1.0 \times 10^{-5}$ Ha, 0.002 Ha/Å, and 0.005 Å, respectively. Additionally, the related parameters of total energy including the global orbital cut-off radius and smearing were set as 5.2 Å and 0.005 Ha, respectively. To prevent interactions between adjacent units, we adopted a vacuum layer of 20 Å in this work [40].

The adsorption energy ($E_{ads}$) of the gas molecules on the pristine Pc and TMPc monolayers was determined to assess the gas capture ability as in Equation (1), as follows [41,42]:

$$E_{ads} = E_{gas+TMPc} - E_{TMPc} - E_{gas} \tag{1}$$

where $E_{gas+TMPc}$ is the total energy of the TM-doped phthalocyanine with gas adsorption, and $E_{TMPc}$ and $E_{gas}$ are the total energies of the TMPc monolayer and gas molecule, respectively. Generally, a negative $E_{ads}$ value indicates that adsorption is an exothermic process and can occur spontaneously; a more negative value indicates better stability.

The CDD of different adsorption systems was calculated to visualize the charge transfer ($Q_t$) between gas molecules and TMPc monolayers. Estimating $Q_t$ based on the Hirshfeld charge is defined by Equation (2), as follows [43]:

$$Q_t = Q_{adsorbed\ molecule} - Q_{isolated\ molecule} \tag{2}$$

where $Q_{\text{adsorbed molecule}}$ and $Q_{\text{isolated molecule}}$ indicate the charge numbers of the target molecules before and after adsorption, respectively. $Q_t > 0$ indicates that the electrons are transferred from the gas molecule to the TMPc surface; otherwise, the electrons are transferred from the TMPc surface to the gas molecule.

## 3. Results and Discussion

### 3.1. Optimum Structure and Stable Adsorption Configuration

Figure 1 depicts the fully-relaxed structures of Pc, PdPc, RuPc, and $CH_2O$ gas molecules. Figure 1a shows the molecular structure of an unsubstituted Pc nanosheet, which contains 20 C, 8 N, and 4 H atoms. All the atoms within the Pc molecule are coplanar and fully conjugated. Different metal phthalocyanine complexes have different structures [44], and according to the investigation conducted by Mellor and Maley [45], the bivalent Pd ion in the complex has the absolute stability. Moreover, ruthenium coordination complexes exhibit a wide range of formal metal oxidation states [46], and the majority of reported RuPc complexes have Ru(II) metal centers. Therefore, in this work, the PdPc and RuPc nanosheet is a monolayer of metal phthalocyanine complex formed by decorating $Pd^{2+}$ and $Ru^{2+}$ ions into the central cavity of the phthalocyanine, as displayed in Figure 1b,c. These two metal-embedded phthalocyanine monolayers remain "graphene-like" plane configurations, and their lattice parameters have a negligible change. The calculated lattice constant of the TMPc monolayer is about 10.69 Å, which agrees well with the experimental value [47,48] and other theoretical calculations [49,50]. Figure 1d shows the optimized structure of the $CH_2O$ gas molecule. In the optimized $CH_2O$ geometric model, the bond lengths of C-O and C-H are calculated to be 1.214 Å and 1.118 Å, respectively, and the bond angle (H-C-O) is found to be 121.931°, which is consistent with experimental value and other theoretical studies [51].

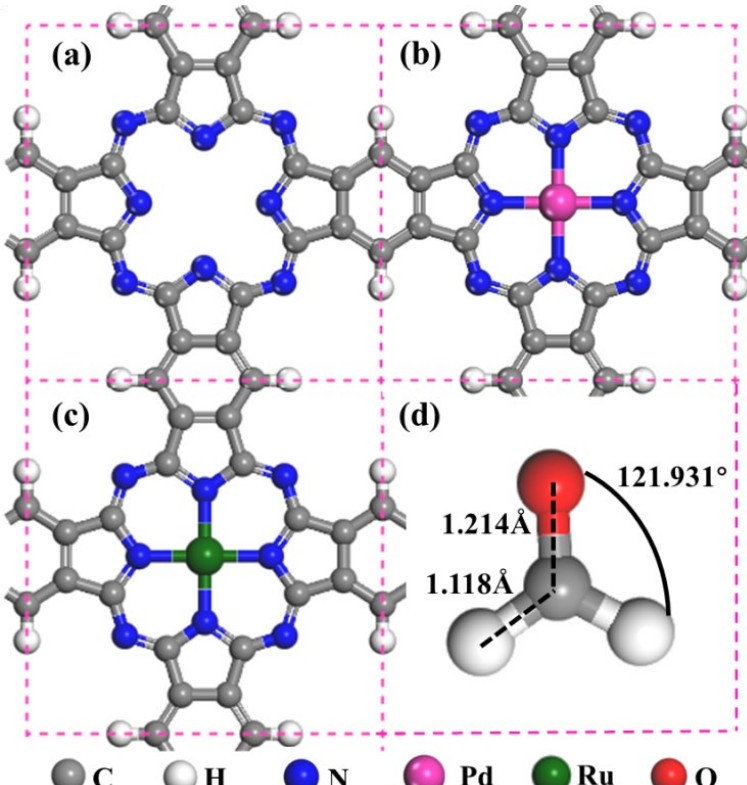

**Figure 1.** Optimized structure model: (**a**) Pc monolayer, (**b**) PdPc monolayer, (**c**) RuPc monolayer, and (**d**) $CH_2O$ gas molecule.

Figure 2 shows the diagram of binding energy ($E_{bin}$) for the TM atoms embedded in the Pc sheet. It was found that the binding energy ($E_{bin}$) between doped TM atoms and Pc nanosheet was in the range of $-8.23$ eV to $-10.11$ eV, and all the values were lower than the corresponding metal bulk cohesive energy ($E_{coh}$). This outcome demonstrates the high thermodynamic stability of the PdPc and RuPc monolayers.

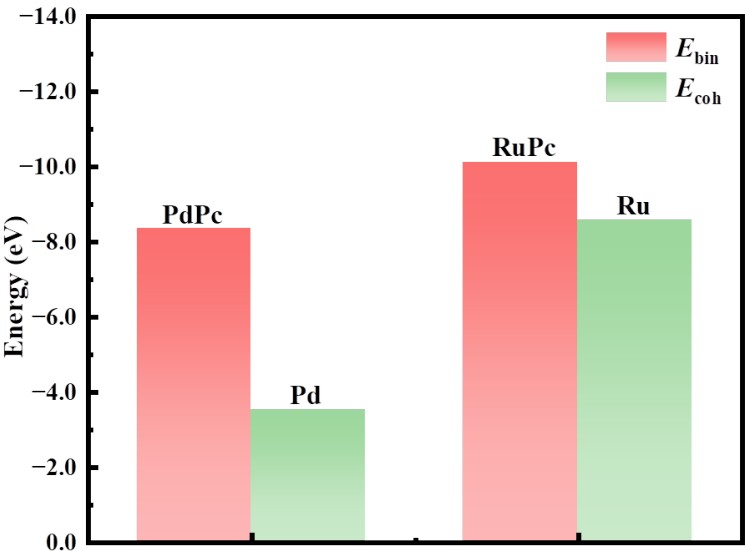

**Figure 2.** Binding energy ($E_{bin}$) between Pd, Ru, and Pc monolayers and cohesive energy ($E_{coh}$) of Pd and Ru bulk.

Figure 3 depicts the charge-density difference (CDD) and density of states (DOSs) of the PdPc and RuPc monolayers. Figure 3a,c show electron-depletion regions around Pd and Ru atoms, whereas many electrons accumulate around N and C atoms of Pc molecules. This result indicates that there exists a strong covalent bond between the TM and Pc monolayers. As displayed in Figure 3b,d, the Pd-d and Ru-d orbital is greatly hybrid with Pc orbital at the range of $-10.0$ to 7.5 eV, which shows that the interaction between Pd and Ru atoms and Pc is strong. Consequently, the Pd and Ru atoms have a strong affinity with Pc monolayers. This outcome also further demonstrates the structural stability of the PdPc and RuPc nanosheets, which is consistent with the calculated results of binding energy ($E_{bin}$) displayed in Figure 2.

### 3.2. Adsorption Properties of CH2O on Pc, PdPc, and RuPc Monolayers

According to our previous investigation [52], the Pc nanosheet central vacancy or just above the central metal atom of the TMPc monolayer was regarded as the gas molecule preferential adsorption site. In this work, we also considered two different adsorption styles, namely, $CH_2O$ gas molecule adsorbed on the Pc nanosheet in parallel and vertical styles, which we defined as $CH_2O@Pc$-P and $CH_2O@Pc$-V, respectively, as shown in Figure 4($a_1$,$b_1$). For the first adsorption style ($CH_2O@Pc$-P), after $CH_2O$ adsorption on the Pc monolayer, one can see that the gas molecule prefers to adsorb on the Pc nanosheet with an oriented H atom and the planar Pc slightly bends at the adsorption site towards the $CH_2O$ gas, as shown in Figure 4($a_2$). The corresponding adsorption energy ($E_{ads}$) was found to be $-0.58$ eV, and an amount of 0.0163 e was transferred from the Pc nanosheet to the $CH_2O$ gas molecule, as seen in Table 1. Nevertheless, after the $CH_2O$ gas molecule adsorption on the Pc nanosheet in vertical style, as shown in Figure 4($b_2$), it was found that the two H atoms of gas had been dissociated and only the O and C atoms remained in the gaseous phase. The related adsorption energy ($E_{ads}$) of the $CH_2O@Pc$-V adsorption system was $-4.27$ eV and the large amount of 0.1969 e was transferred from the gas molecule to the Pc nanosheet (Table 1). Although the pristine Pc had a noticeable change in the conducting behavior after the $CH_2O$ gas molecule adsorption in vertical style (seeing the change of

band gap in Table 1), the ultra-strong adsorption strength restricted its further gas-sensing application [52].

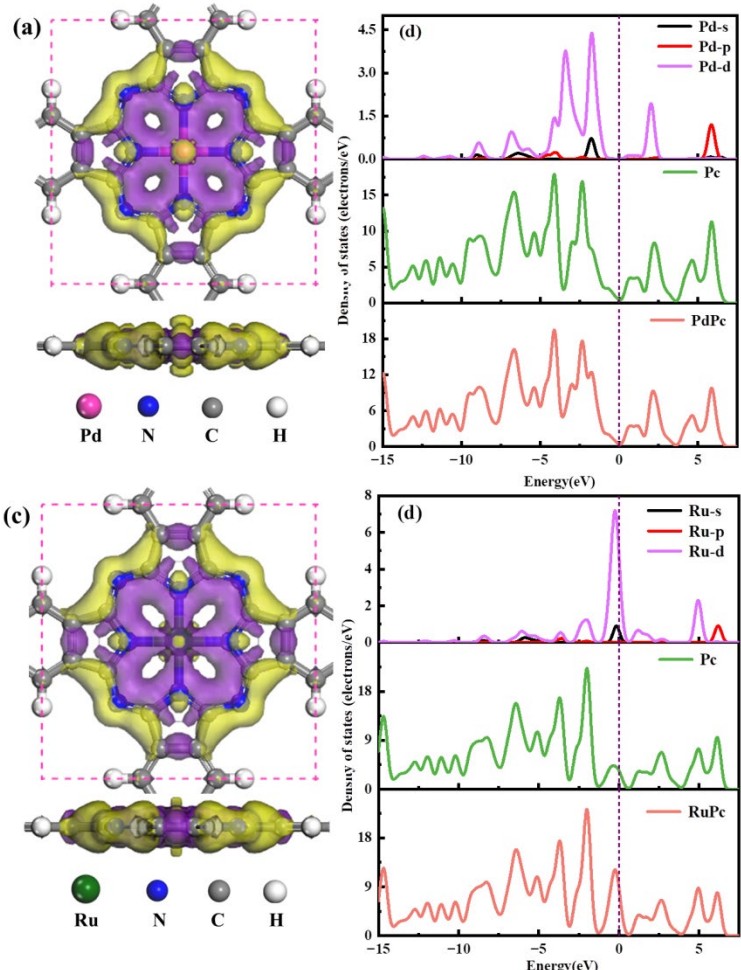

**Figure 3.** Charge-density difference (CDD) and density of states (DOSs) of the (**a**,**b**) PdPc and (**c**,**d**) RuPc monolayers. The yellow (purple) region represents electron accumulation (depletion), and the isosurface value of CDD is $\pm 0.02$ e/$\text{Å}^3$. The Fermi level is set to zero energy and indicated by the vertical purple dashed line.

To further explore a potential candidate as a gas-sensing material for $CH_2O$ gas detection, we selected the Pd and Ru dopant atoms to tune the adsorption performance of pristine Pc. Table 2 lists the detailed calculated information, including the adsorption energy, adsorption distance, charge transfer, and band gap for the $CH_2O$@PdPc and $CH_2O$@RuPc adsorption systems. It is found that the $CH_2O$ molecule preferred to adsorb on PdPc and RuPc sheets in parallel style, and the corresponding adsorption energies were $-0.22$ eV and $-0.91$ eV, respectively. This outcome shows that the adsorption of the $CH_2O$ molecule on the PdPc sheet is physisorption, which primarily depends on the van der Waals force. By contrast, the RuPc monolayer has a stronger affinity with the $CH_2O$ gas molecule. Additionally, there is a noticeable change in the conducting behavior of RuPc after the $CH_2O$ gas molecule adsorption, demonstrating that the RuPc sheet is highly sensitive to these gas molecules.

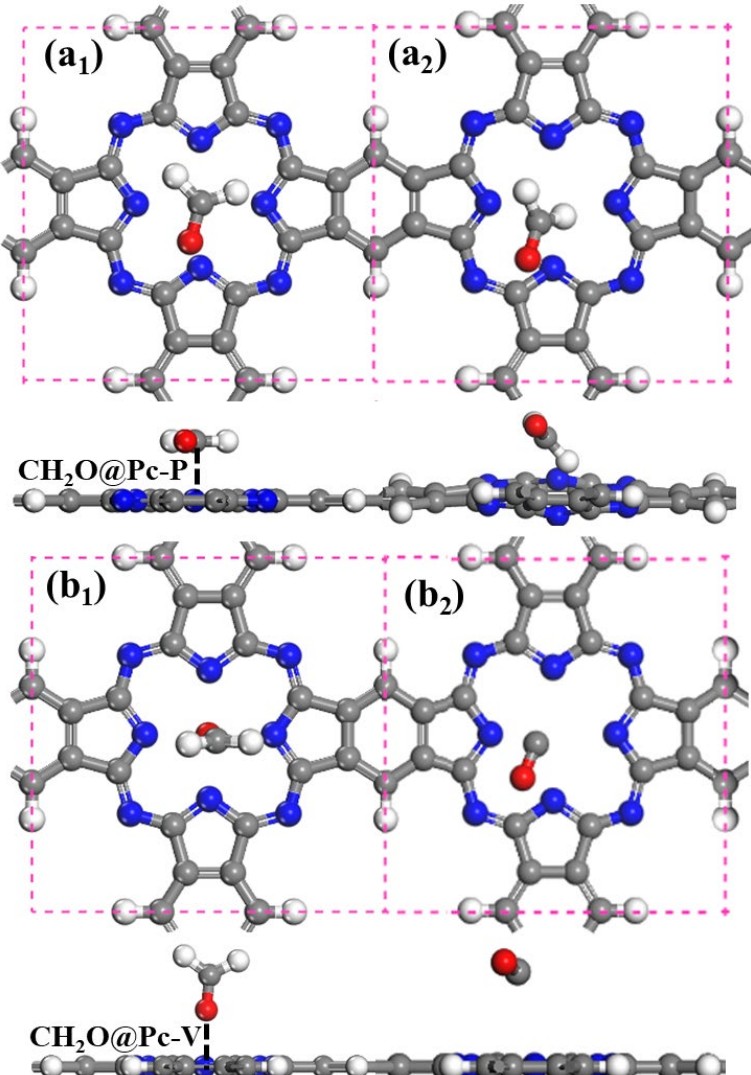

**Figure 4.** Top and side views of original structure before adsorption on Pc monolayer: (**a₁**) CH₂O-P and (**b₁**) CH₂O-V, and the optimized structure of different adsorption systems (**a₂**) CH₂O@Pc-P and (**b₂**) CH₂O@Pc-V.

**Table 1.** The adsorption energy ($E_{ads}$), charge transfer ($Q_t$), and band gap of the system before adsorption ($E_g$) and after adsorption ($E_g'$) for the CH₂O@Pc adsorption system.

| Gas Molecule | Material | Adsorption Style | $E_{ads}$/eV | $Q_t$/e | $E_g$ ($E_g'$)/eV |
|:---:|:---:|:---:|:---:|:---:|:---:|
| CH₂O | Pc | Parallel | −0.58 | −0.0163 | 0.00 (0.00) |
| | | Vertical | −4.27 | 0.1969 | 0.00 (1.12) |

**Table 2.** The adsorption energy ($E_{ads}$), adsorption distances ($D$), charge transfer ($Q_t$), and band gap of the system before adsorption ($E_g$) and after adsorption ($E_g'$) of CH₂O@PdPc and CH₂O@RuPc adsorption systems.

| Gas Molecule | Material | Adsorption Style | $E_{ads}$/eV | $d_{M-O}$/Å | $d_{M-C}$/Å | $Q_t$/e | $E_g$ ($E_g'$)/eV |
|:---:|:---:|:---:|:---:|:---:|:---:|:---:|:---:|
| CH₂O | PdPc | Parallel | −0.22 | 3.560 | 3.020 | −0.0929 | 0.00 (0.00) |
| | | Vertical | −0.18 | 3.222 | 3.781 | −0.0189 | 0.00 (0.00) |
| | RuPc | Parallel | −0.91 | 2.058 | 2.096 | −0.1312 | 0.00 (0.853) |
| | | Vertical | −0.89 | 1.974 | 2.874 | 0.1321 | 0.00 (0.792) |

Figure 5 illustrates the lowest-energy configurations along with charge-density difference (CDD) and electron-density distribution (EDD) of $CH_2O$ adsorbed on PdPc and RuPc monolayers. In Figure 5($a_1$), one can see that the $CH_2O$ gas molecule adsorbs at a large distance of about 3.560 Å on the surface of the PdPc nanosheet with lower adsorption energy of −0.22 eV. This interaction reveals weak physisorption, primarily owing to the interaction of the van der Waals force. Figure 5($a_2$,$a_3$) displays the CDD and EDD maps of the $CH_2O$@PdPc adsorption system, it is seen that there is no distinct electron-density overlap between the $CH_2O$ gas molecule and the PdPc sheet, and a very small amount of charge is transferred from the $CH_2O$ gas molecule to the PdPc sheet due to its weak interaction. Compared with the case of the PdPc sheet, the $CH_2O$ gas molecule experienced a stronger interaction, the corresponding adsorption energy was found to be −0.91 eV with a smaller adsorption distance of 2.096 Å, as shown in Figure 5($b_1$). From the CDD maps, as shown in Figure 5($b_2$), one can see an evident electron-depletion region around the Ru atom, whereas many electrons were accumulated around the $CH_2O$ gas and Pc molecule. This outcome suggests that the doped Ru behaves as a bridge enhancing the interaction between the $CH_2O$ gas and Pc nanosheet. Moreover, the EDD diagram of the $CH_2O$@RuPc adsorption system (shown in Figure 5($b_3$)) also shows the strong overlapping of charge density between the $CH_2O$ gas molecule and RuPc sheet. Consequently, the RuPc monolayer has stronger capture ability towards $CH_2O$ gas than the PdPc one.

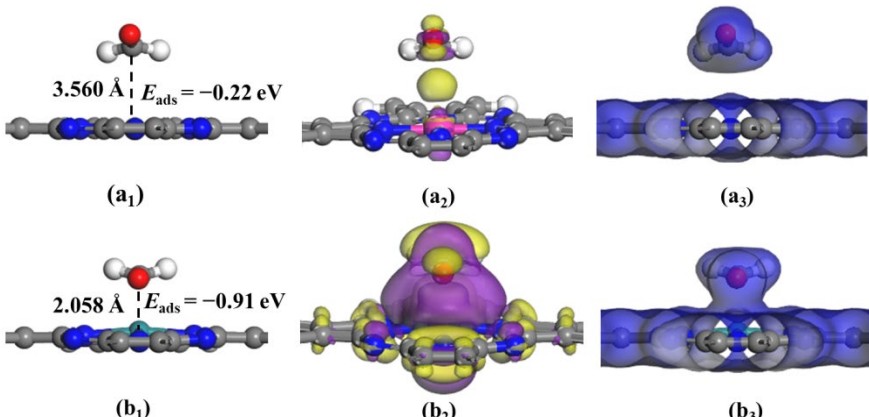

**Figure 5.** Lowest-energy structures, charge-density difference (CDD), and electron-density distribution (EDD) of $CH_2O$ adsorbed on PdPc and RuPc monolayers. ($a_1$–$a_3$) represent the $CH_2O$@PdPc adsorption system and ($b_1$–$b_3$) represent the $CH_2O$@RuPc adsorption system. The yellow (purple) region represents electron accumulation (depletion), and the isosurface of CDD and EDD is ±0.005 e/Å³ and 0.2 e/Å³, respectively.

To further study the electronic behaviors of $CH_2O$@PdPc and $CH_2O$@RuPc adsorption systems for understanding gas-sensing mechanisms, the corresponding DOSs and PDOSs spectra of these two adsorption systems are displayed in Figure 6. From Figure 6a, it can be seen that after the adsorption of $CH_2O$ gas on the PdPc sheet, the DOSs had no obvious shift, and remained almost the same; that is, the $CH_2O$ gas does not influence the electronic properties of the PdPc nanosheet. Moreover, no prominent resonance peaks were observed between O-p and Ru-d orbitals, as shown in Figure 6c. This result further implies that the adsorption interaction of $CH_2O$ on the PdPc monolayer is mainly due to the van der Waals force. In Figure 6b, the DOSs spectra of the $CH_2O$@RuPc adsorption system and the pristine RuPc in the range of −15 eV to 7.5 eV has obvious deviation. The dominant peak of RuPc around the Fermi level disappears after the adsorption of $CH_2O$ gas. Hence, the $CH_2O$ gas molecule can significantly affect the electronic properties of the RuPc nanosheet. From the PDOSs displayed in Figure 6d, one can see that their strong adsorption strength is primarily attributed to the great orbital hybridization between O-p and Ru-d, and numerous dominant resonance peaks are found at the range of −7.5 to 2.5 eV.

This outcome further shows the adsorption energy ($E_{ads}$) absolute value and configuration of the $CH_2O$@RuPc adsorption system is high and stable, respectively.

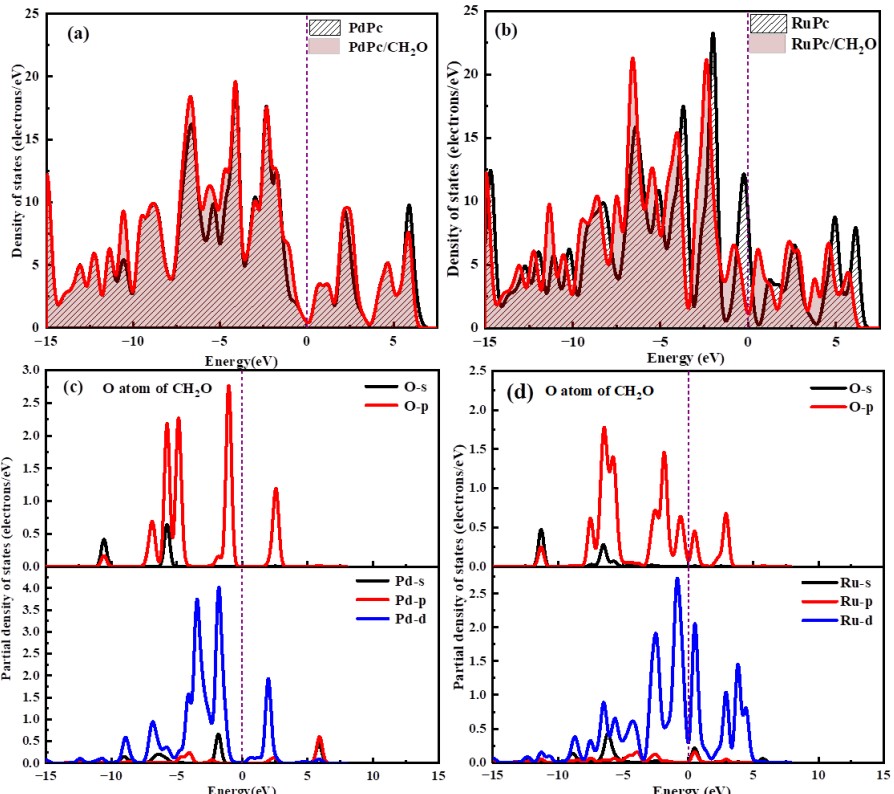

**Figure 6.** The density of states (DOSs) and partial density of states (PDOSs) of $CH_2O$@PdPc and $CH_2O$@RuPc adsorption systems. (**a**,**b**) DOSs map before and after $CH_2O$ adsorption on PdPc and RuPc, and (**c**,**d**) PDOSs of O, Pd, and Ru atoms. The Fermi level is set to zero energy and indicated by the purple dashed line.

### 3.3. Sensing Performance Evaluation of PdPc and RuPc

Generally, the variation in the electrical conductivity ($\sigma$) is a critical factor for assessing gas-sensor sensitivity. Therefore, the sensing performance of PdPc and RuPc towards $CH_2O$ gas molecules was evaluated by evaluating the electrical conductivity's variation. The $\sigma$ of a system is defined as Equation (3) as follows [53]:

$$\sigma \propto \exp\left(-E_g / 2K_B T\right) \tag{3}$$

where the $\sigma$ is the adsorption system's electric conductivity, $E_g$ is the band gap, and $K_B$ and $T$ are the Boltzmann constant ($8.62 \times 10^{-5}$ eV/K) and temperature, respectively.

Figure 7 shows the band structures of different adsorption systems. From Figure 7a,b, for the $CH_2O$@PdPc adsorption system, one can see that the band gap of the PdPc nanosheet remained the same after the $CH_2O$ adsorption, which indicates that the $CH_2O$ gas molecule does not affect the electrical conductivity of the PdPc monolayer. This outcome suggests that the PdPc cannot be a candidate gas-sensing material for $CH_2O$ owing to its poor sensitivity. Interestingly, after $CH_2O$ gas adsorption, an obvious band gap variation (up from 0 to 0.853 eV) of the RuPc monolayer can be observed in Figure 7c,d, which shows that the conducting feature of the RuPc nanosheet is changed into a semiconducting feature. In other words, an Ru-decorated Pc monolayer can be used as gas-sensing material with high sensitivity to $CH_2O$.

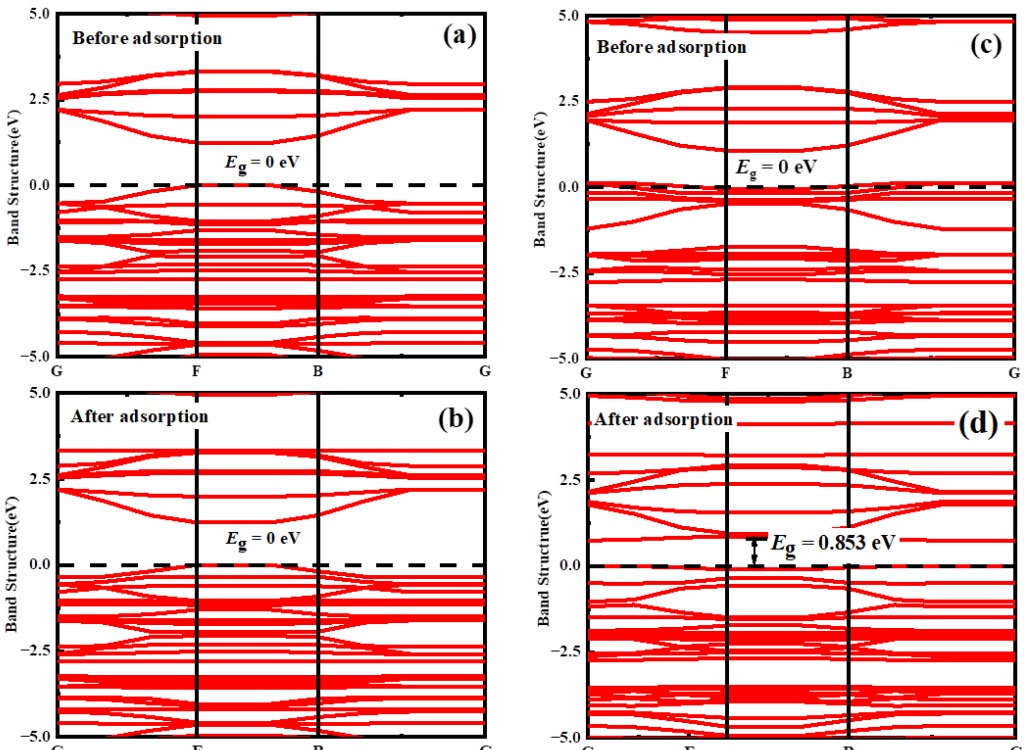

**Figure 7.** Band structures of different adsorption systems, (**a,b**) CH$_2$O@PdPc and (**c,d**) CH$_2$O@RuPc.

More specifically, the sensitivity (*S*) of the CH$_2$O gas sensor was evaluated by the resistance variation of CH$_2$O@RuPc adsorption systems. It is well-known that the resistance of a material is inversely proportional to its conductivity. When incorporating conductivity into Equation (3), the sensor sensitivity can be calculated by the following Equation [53]:

$$S = \left( \frac{1}{\sigma_{\mathrm{CH_2O@RuPc}}} - \frac{1}{\sigma_{\mathrm{RuPc}}} \right) / \frac{1}{\sigma_{\mathrm{RuPc}}} = \exp\left[ (E_{\mathrm{g\ CH_2O@RuPc}} - E_{\mathrm{g\ RuPc}})/2K_{\mathrm{B}}T \right] - 1 \quad (4)$$

where the $\sigma_{\mathrm{CH_2O@RuPc}}$ and $\sigma_{\mathrm{RuPc}}$ represent the conductivity of the RuPc monolayer after and before adsorption.

Figure 8 shows the sensitivity (*S*) for the CH$_2$O gas on the RuPc monolayer. Obviously, within the normal working temperature range, the sensitivity at 298 K, 348 K, and 398 K was $1.65 \times 10^7$, $1.51 \times 10^6$, and $2.53 \times 10^5$, respectively. Therefore, the sensitivity of the RuPc monolayer is sufficiently high for CH$_2$O gas-sensing.

Finally, according to the transition state theory and the Van't Hoff–Arrhenius formula [54], the recovery time ($\tau$) can be expressed by the following equation [54]:

$$\tau = v_0^{-1} \exp\left( \frac{-E_{\mathrm{ads}}}{K_{\mathrm{B}}T} \right) \quad (5)$$

Here, the attempt frequency ($v_0$) is $10^{-12}$ s$^{-1}$.

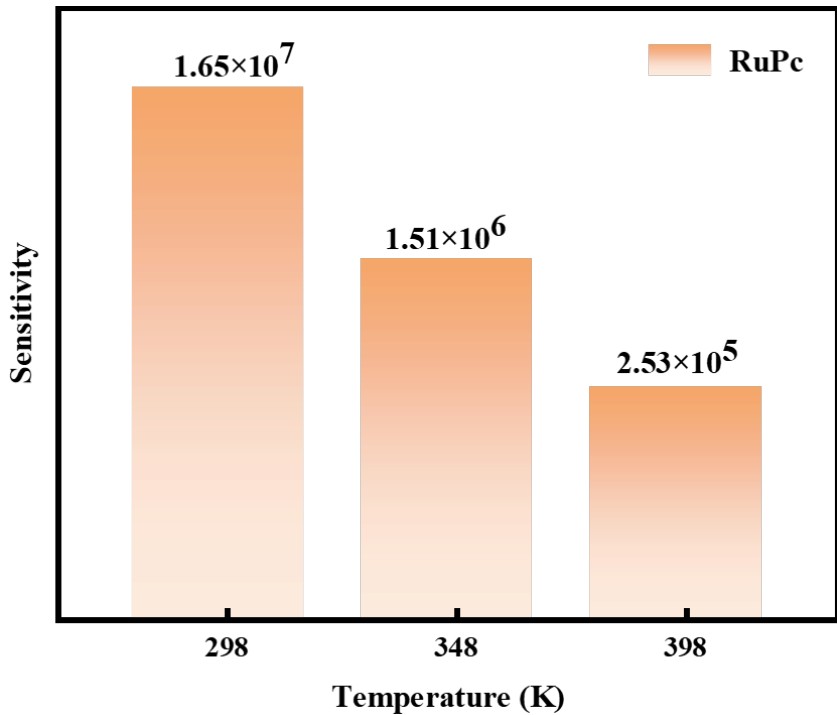

**Figure 8.** Sensitivity of RuPc monolayer towards the CH$_2$O gas at different temperatures.

Figure 9 depicts the predicted recovery time of the CH$_2$O gas molecule at different temperatures (298, 348, and 398 K). Although the PdPc monolayer has a fast desorption speed, it cannot be a CH$_2$O gas-sensing material owing to its weak gas-capture ability. For the RuPc nanosheet, one can see that the recovery time of the CH$_2$O gas molecule at room temperature (298 K) was 2427.54 s. With the temperature increased to 348 K and 398 K, the corresponding recovery time is reduced to 14.95 s and 0.33 s, respectively. The comparative information of different 2D materials towards CH$_2$O-sensing with previously reported literature is summarized in Table 3. Firstly, one can see that the adsorption of CH$_2$O on the graphene g-C$_3$N$_4$ sheet [55], Ti$_3$C$_2$O$_2$ MXene [56], and SnS monolayers [57] is physical adsorption, and their interaction mainly depends on the van der Waals force. Hence, these 2D materials are not suitable to be a gas-sensing material for CH$_2$O owing to the weak gas-capture ability. In contrast, the Li-doped ethylene [58], Al-doped C$_2$N sheet [59], and Si-doped BC$_3$ sheet [60] have ultra-strong adsorption strength for CH$_2$O gas molecules, which makes desorption process challenging due to their ultra-long recovery time. Therefore, they are also not suitable for reusable CH$_2$O gas-sensing materials. In conclusion, by considering the capture ability and recovery time, the RuPc monolayer can be regarded as a recyclable gas-sensing material for CH$_2$O gas detection.

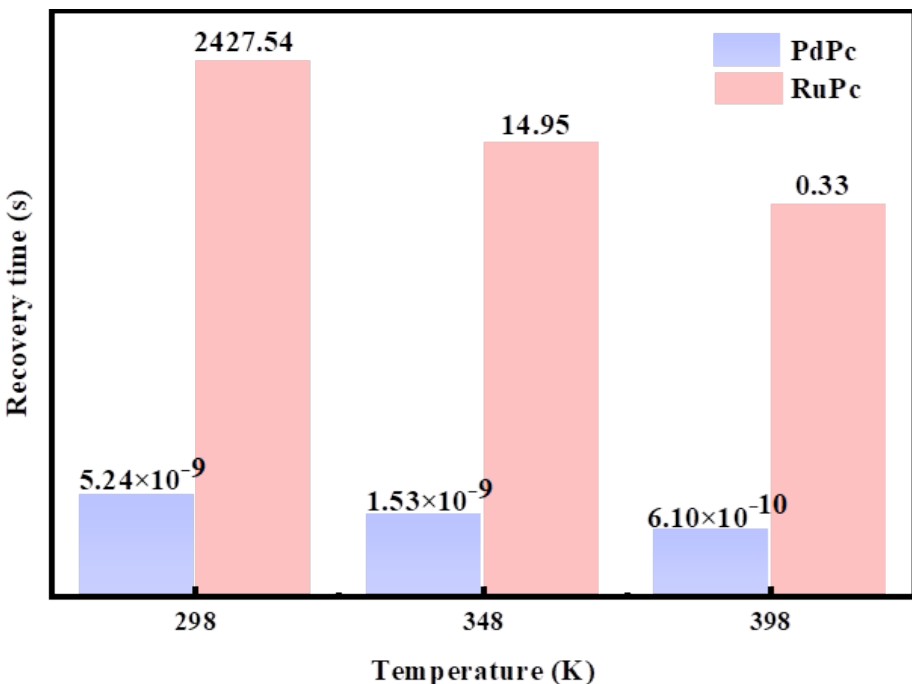

**Figure 9.** Recovery time of PdPc and RuPc towards the $CH_2O$ at different temperatures.

**Table 3.** Comparison of different properties for $CH_2O$ adsorbed on different 2D materials.

| 2D Materials | $E_{ads}$ (eV) | $Q_t$ (e) | $\Delta E_g$ (%) | $\tau$ (s, 298 K) | Refs |
|---|---|---|---|---|---|
| Graphene g-$C_3N_4$ sheet | −0.317 | | | $2.29 \times 10^{-7}$ | [55] |
| $Ti_3C_2O_2$ MXene | −0.300 | | | $1.18 \times 10^{-7}$ | [56] |
| SnS monolayers | −0.300 | −0.110 | | $1.18 \times 10^{-7}$ | [57] |
| Li-doped ethylene | −1.310 | | 11.0 | $1.41 \times 10^{10}$ | [58] |
| Al-doped $C_2N$ sheet | −2.754 | −0.438 | | $3.64 \times 10^{34}$ | [59] |
| Si-doped $BC_3$ sheet | −1.149 | −0.130 | 28.2 | $2.67 \times 10^7$ | [60] |
| RuPc monolayer | −0.910 | −0.131 | 100 | 2427 | This work |

## 4. Conclusions

This study used first-principles calculations to investigate the adsorption behaviors of toxic $CH_2O$ gas molecules on TMPc (TM = Pd, Ru) monolayers. The adsorption configurations and electronic structures of different adsorption systems were systematically analyzed, and the corresponding sensing mechanism was revealed. The main conclusions are as follows:

(1) Transition-metal Pd and Ru atoms can significantly alter the electronic properties of Pc and the calculation result proves that these two TMPc monolayers are thermodynamically stable;

(2) The RuPc monolayer possesses a stronger affinity with the $CH_2O$ gas molecule because of the strong orbital hybridization between Ru-d and O-p of the gas molecule;

(3) The RuPc monolayer is highly sensitive to $CH_2O$ and it can be regarded as a potential gas-sensing material for $CH_2O$ detection owing to its suitable adsorption ability and desorption time.

Overall, our DFT calculations can guide experimentalists in performing relevant research on TMPc monolayers for future gas-sensing applications of $CH_2O$ gas molecules.

**Author Contributions:** Conceptualization, Z.N. and R.X.; methodology, R.X. and C.W.; visualization, Z.N.; software, Z.N. and E.D.; validation, C.W.; investigation, R.X. and C.W.; data curation, Y.W. and Q.G.; writing—original draft, R.X. and Z.N.; writing—review and editing, Z.N. and E.D. All authors have read and agreed to the published version of the manuscript.

**Funding:** This work was supported by the National Natural Science Foundation of China (Grant No. 51904137), the Special Basic Cooperative Research Programs of Yunnan Provincial Undergraduate Universities' Association (Grant No. 202101BA070001-032), the Basic Research Project of Yunnan Province (202201AT070017), the High-Level Talent Plans for Young Top-notch Talents of Yunnan Province (Grant No. YNWR-QNBJ-2020-017), and the High-Level Talent Special Support Plans for Young Talents of Kunming City (Grant No. C201905002).

**Institutional Review Board Statement:** Not applicable.

**Informed Consent Statement:** Not applicable.

**Data Availability Statement:** The data presented in this study are contained within the article.

**Acknowledgments:** We thank the associate editor and the reviewers for their useful help in improving this paper, along with the Scientific Innovation Team of Kunming University for helpful discussions on topics related to this work. The authors wish to acknowledge Lei Gao, Kunming University of Science and Technology, for his help in editing of English language and style.

**Conflicts of Interest:** The authors declare no conflict of interest.

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
