# Peer review of "Metal Embedded Phthalocyanine Monolayers as Promising Materials for Toxic Formaldehyde Gas Detection: Insights from DFT Calculations"

_metals, doi:10.3390/met12091442_

Round 1
Reviewer 1 Report
The manuscript is complete in every part. The authors used simple and clear language. The results proved almost torally the objectives described.
The work can be accepted in the form presented.
Author Response
Journal: Metals
Manuscript ID: metals-1874186
Title: Metal embedded phthalocyanine monolayers as promising materials for CH2O toxic gas detection: Insights from DFT calculations
Author(s): Xue Rou, Wang, Chen; Wang, Yajun; Guo, Qijun; Dai, Enrui; Nie, Zhifeng*
List of Responses
Dear Editors and Reviewers:
Thank you for your letter and for the reviewers’ comments concerning our manuscript entitled “Metal embedded phthalocyanine monolayers as promising materials for CH2O toxic gas detection: Insights from DFT calculations” (Manuscript ID: metals-1874186). Those comments are all valuable and very helpful for revising and improving our paper, as well as the important guiding significance to our researches. We have studied comments carefully and have made correction which we hope meet with approval. Revision have been made and highlighted with red words in the revised manuscript. The main corrections in the paper and the responds to the reviewer’s comments are as flowing:
Responds to the reviewer’s comments:
Reviewer #1:
Recommendation: The work can be accepted in the form presented.
Response: Thanks a lot for the reviewer’s careful review. We appreciate the reviewer’s comments. We tried our best to improve the manuscript and made some changes in the revised manuscript. Those changes will not influence the content and framework of the paper, and hope that the correction will meet with approval.
Thanks again for your kind consideration
Yours sincerely,
Zhifeng Nie,
Yunnan Key Laboratory of Metal-Organic Molecular Materials and Device, Kunming University, Kunming 650214, China
E-mail: niezf123@163.com

Reviewer 2 Report
In my opinion, the paper can be published after making some minor revisions and some improvements in the presentation of the article, which are as follows:
1. The abstract does not truly represent the paper. Rephrase the Abstract to become more concise, comprehensible and clear.
2. The novelty of the work should be established
3. Equations 1 and 2 need references.
4. Why you did not use other more powerful theoretical?
Thus, the manuscript should experience the minor revision before acceptance.
Author Response
Journal: Metals
Manuscript ID: metals-1874186
Title: Metal embedded phthalocyanine monolayers as promising materials for CH2O toxic gas detection: Insights from DFT calculations
Author(s): Xue Rou, Wang, Chen; Wang, Yajun; Guo, Qijun; Dai, Enrui; Nie, Zhifeng*
List of Responses
Dear Editors and Reviewers:
Thank you for your letter and for the reviewers’ comments concerning our manuscript entitled “Metal embedded phthalocyanine monolayers as promising materials for CH2O toxic gas detection: Insights from DFT calculations” (Manuscript ID: metals-1874186). Those comments are all valuable and very helpful for revising and improving our paper, as well as the important guiding significance to our researches. We have studied comments carefully and have made correction which we hope meet with approval. Revision have been made and highlighted with red words in the revised manuscript. The main corrections in the paper and the responds to the reviewer’s comments are as flowing:
Responds to the reviewer’s comments:
Reviewer #2:
Recommendation: In my opinion, the paper can be published after making some minor revisions and some improvements in the presentation of the article. Thus, the manuscript should experience the minor revision before acceptance.
Comment 1: The abstract does not truly represent the paper. Rephrase the Abstract to become more concise, comprehensible and clear.
Response: Thanks a lot for this valuable suggestion. According to your comment, we have rephrased our abstract, which is as follows: Design of the good performance materials for the toxic formaldehyde (CH2O) gas is critical for environmental preservation and human health. In this work, density functional theory (DFT) calculations were employed to investigate the adsorption behavior and electronic properties of CH2O on the transition metal (TM) doped phthalocyanine monolayers. Our results prove that PdPc and RuPc monolayers are thermodynamically stable. The analysis of the adsorption energy shows that the CH2O gas molecule is chemisorbed on the RuPc monolayer, while it is physisorbed on the PdPc nanosheet. The microcosmic interaction mechanism within the gas–adsorbent system is revealed by analyzing the density of states, charge density difference, electron density distribution and Hirshfeld charge transfer. Additionally, the RuPc monolayer is highly sensitive to CH2O due to the obvious changes in electrical conductivity, and the recovery time of CH2O molecule is predicted to be 2427 s at room temperature. Therefore, RuPc monolayer can be regarded as a promising gas sensing material for CH2O detection.
Comment 2: The novelty of the work should be established.
Response: Thanks a lot for your careful review. According to your advices, we have rephrased our manuscripts and established the novelty of the work. The highlights of this work were summarized as following:
- Pd and Ru atoms can significantly alter the electronic properties of Pc monolayer.
- RuPc monolayer shows appropriate capture ability for CH2O molecule due to the hybridizations between Ru-d and O-p orbitals.
- RuPc monolayer is highly sensitive to CH2O gas molecule.
- RuPc monolayer can be regarded as a promising material for CH2O detection.
Comment 3: Equations1 and 2 need references.
Response: Thanks a lot for your careful review. According to your advices, we have added the corresponding references for Equations1 and 2, as following:
- M. Jabrane, M. El Hafidi, M.Y. El Hafidi, A. Kara, Fe-Phthalocyanine on Cu(111) and Ag(111): A DFT+vdWs investigation, Surface Science, 716 (2022) 121961.
- E.V. Basiuk, L. Huerta, V.A. Basiuk, Noncovalent bonding of 3d metal(II) phthalocyanines with single-walled carbon nanotubes: A combined DFT and XPS study, Applied Surface Science, 470 (2019) 622-630.
- S.U.D. Shamim, D. Roy, S. Alam, A.A. Piya, M.S. Rahman, M.K. Hossain, F. Ahmed, Doubly doped graphene as gas sensing materials for oxygen-containing gas molecules: A first-principles investigation, Applied Surface Science, 596 (2022) 153603.
Comment 4: Why you did not use other more powerful theoretical?
Response: Thanks a lot for this valuable suggestion. Massive researchers have conducted various theoretical investigations on the 2D sensing material, and the DFT calculation is confirmed as an effective tool to investigate the adsorption behavior and electronic properties of gas molecules on the related 2D metal embedded phthalocyanine monolayers, as shown in the following references.
- Alosabi, A. Q.; Al-Muntaser, A. A.; El-Nahass, M. M.; Oraby, A. H. Structural, optical and DFT studies of disodium phthalocyanine thin films for optoelectronic devices applications. Opt. Laser. Technol. 2022, 155, 108372.
- Zhou, Y.; Gao, G.; Chu, W.; Wang, L. W. Computational screening of transition metal-doped phthalocyanine monolayers for oxygen evolution and reduction. Nanoscale. Adv. 2022, 2, 710-716.
- M. Jabrane, M. El Hafidi, M.Y. El Hafidi, A. Kara, Fe-Phthalocyanine on Cu(111) and Ag(111): A DFT+vdWs investigation, Surface Science, 716 (2022) 121961.
- E.V. Basiuk, L. Huerta, V.A. Basiuk, Noncovalent bonding of 3d metal(II) phthalocyanines with single-walled carbon nanotubes: A combined DFT and XPS study, Applied Surface Science, 470 (2019) 622-630.
We tried our best to improve the manuscript and made the corresponding changes in the manuscript. We appreciate for your warm work earnestly, and hope that the correction will meet with approval. Once again, thank you very much for your comments and suggestions.
Thanks again for your kind consideration
Yours sincerely,
Zhifeng Nie,
Yunnan Key Laboratory of Metal-Organic Molecular Materials and Device, Kunming University, Kunming 650214, China
E-mail: niezf123@163.com

Reviewer 3 Report
This manuscript presents the results of investigations on a relevant subject matter of Journal «Metals». An attempt was made to theoretically substantiate the interaction of formaldehyde with palladium and ruthenium phthalocyanines. In my opinion, an interesting problem is being discussed, but some important factors are not taken into account.
Comments
Important remarks
(I) What is the oxidation state of palladium in the complex? How was this factor taken into account in the calculations?
(II) What is the oxidation state of ruthenium in the complex? How was this factor taken into account in the calculations?
(III) It is known that in ruthenium phthalocyanine the metal has a coordination number of 6. Why was this fact ignored in the calculations? Could it be that this led to such significant value of energy for this complex?
Minor remarks
(IV) Do not include the chemical formulas of compounds in the title of the manuscript and Keywords.
(V) The manuscript section should not begin with a figure or table.
(VI) Figure 4b2: Why is the interaction of carbon monoxide with phthalocyanine shown in the figure?
(VII) Figures 5, 6: Figure captions should be on the same page as them.
(VIII) The writing needs some revision by a native English speaker.
In general, the topic of the manuscript is relevant, but only data for palladium phthalocyanine can be published. I hope that my comments will be useful to the authors.
Best regards, Reviewer
Author Response
Journal: Metals
Manuscript ID: metals-1874186
Title: Metal embedded phthalocyanine monolayers as promising materials for CH2O toxic gas detection: Insights from DFT calculations
Author(s): Xue Rou, Wang, Chen; Wang, Yajun; Guo, Qijun; Dai, Enrui; Nie, Zhifeng*
List of Responses
Dear Editors and Reviewers:
Thank you for your letter and for the reviewers’ comments concerning our manuscript entitled “Metal embedded phthalocyanine monolayers as promising materials for CH2O toxic gas detection: Insights from DFT calculations” (Manuscript ID: metals-1874186). Those comments are all valuable and very helpful for revising and improving our paper, as well as the important guiding significance to our researches. We have studied comments carefully and have made correction which we hope meet with approval. Revision have been made and highlighted with red words in the revised manuscript. The main corrections in the paper and the responds to the reviewer’s comments are as flowing:
Responds to the reviewer’s comments:
Reviewer #3:
Recommendation: This manuscript presents the results of investigations on a relevant subject matter of Journal 《Metals》. An attempt was made to theoretically substantiate the interaction of formaldehyde with palladium and ruthenium phthalocyanines. In my opinion, an interesting problem is being discussed, but some important factors are not taken into account. In general, the topic of the manuscript is relevant. But only data for palladium phthalocyanine can be published. I hope that my comments will be useful to the authors.
Comment 1: What is the oxidation state of palladium in the complex? How was this factor taken into account in the calculation?
Response: Thanks a lot for this valuable suggestion. Firstly, in various metal phthalocyanine complexes, phthalocyanine usually exhibits a negative bivalent state, thus, the oxidation state of palladium in the PdPc complex is positive bivalent. This is confirmed by numerous investigations on palladium phthalocyanine (PdPc), which applied in the field of photocatalytic CO2 reduction in Ref. [1]. The corresponding explanation was added in the revised manuscript for details. Secondly, from the point of view of the stability of the complex, the charge and radius of the ion and the stabilization caused by orbital splitting are important. According to the investigation conducted by Mellor and Maley in Ref. [2], the bivalent Pd ion in the complex has the absolute stability. In our calculation, the bivalent Pd ion in the PdPc complex is taken into account.
[1] Y. Gao, R. Zhang, Z. Xiang, B. Yuan, T. Cui, Y. Gao, Z. Cheng, J. Wu, Y. Qi, Z. Zhang, Theoretical insights into photocatalytic CO2 reduction on Palladium phthalocyanine, Chemical Physics Letters, 803 (2022) 139812.
[2] D.P. Mellor, L. Maley, Order of Stability of Metal Complexes, Nature, 161 (1948) 436-437.
Comment 2: What is the oxidation state of ruthenium in the complex? How was factor taken into account in the calculation?
Response: Thanks a lot for this valuable suggestion. The transition metal ruthenium has a rich and varied chemistry, the numerous ruthenium phthalocyanine and naphthalocyanine complexes have been reported (Figure 1) in Ref. [3]. Ruthenium coordination complexes exhibit a wide range of formal metal oxidation states in Ref. [4]. The majority of reported RuPc complex has Ru(â…¡) metal centres although a number of Ru(III) complexes have been reported in Ref. [5]. According the majority reports, we considered ruthenium phthalocyanine (RuPc) complex has Ru(â…¡) metal centres.
In order to clarify these structures of metal phthalocyanine complexes clearly, we have added the necessary explanation and corresponding references in the section 3.1, as following “Different metal phthalocyanine complexes have different structures [44], according to the investigation conducted by Mellor and Maley [45], the bivalent Pd ion in the complex has the absolute stability. Moreover, ruthenium coordination complexes exhibit a wide range of formal metal oxidation states [46], and the majority of reported RuPc complex has Ru(â…¡) metal centres. Therefore, in this work, the PdPc and RuPc nanosheet is a monolayer of metal phthalocyanine complex formed by decorating Pd2+ and Ru2+ ions into the central cavity of the phthalocyanine, as displayed in Figure 1(b-c).”
[3] T. Rawling, A. McDonagh, Ruthenium phthalocyanine and naphthalocyanine complexes: Synthesis, properties and applications, Coordination Chemistry Reviews, 251 (2007) 1128-1157.
[4] G.J. Leigh, Comprehensive coordination chemistry II From Biology to Nanotechnology, Journal of Organometallic Chemistry, 689 (2004) 2733-2742.
[5] S. Sievertsen, H. Schlehahn, H. Homborg, Darstellung, Eigenschaften und elektronische Raman-Spektren von Bis(chloro)phthalocyaninatoferrat(III), -ruthenat(III) und -osmat(III), Zeitschrift für anorganische und allgemeine Chemie, 619 (1993) 1064-1072.
Comment 3: It is known that in ruthenium phthalcoyanine the metal has a coordination number of 6. Why was this fact ignored in the calculations? Could it be that this led to such significant value of energy for this complex?
Response: Thanks a lot for this valuable suggestion. The application of metallophthalocyanines as gas sensing materials has received some attention. Different metal phthalocyanine complexes have different structures, in this work, the PdPc and RuPc nanosheet is a monolayer of metal phthalocyanine complex formed by decorating Pd2+ and Ru2+ ions into the central cavity of the phthalocyanine. As the method (Fig. 50 and Fig. 51) described in Ref. [6], the respective metal complexes can be formed by the adsorbed free-based porphyrins and phthalocyanines at the solid/vacuum interface were shown to react with metal atoms. It is noted that the RuPc complex can be achieved by the method of on-surface metalation. Additionally, several configurations are plausible for achieving a diamagnetic RuPc species, and a number of ways of coordinating the ligands in the complex are possible. As described in Ref. [7], that the axial attachment of the ligands of the complex to a divalent ruthenium is preferred. Therefore, in our calculations for ruthenium phthalocyanine, we considered a coordination number of 4 rather than 6.
[6] J.M. Gottfried, Surface chemistry of porphyrins and phthalocyanines, Surface Science Reports, 70 (2015) 259-379.
[7] W.-H. Chen, T.-H. Huang, K.E. Rieckhoff, E.M. Voigt, Electronic spectra and Zeeman effect of ruthenium phthalocyanine in Shpol'skii matrices, Molecular Physics, 68 (1989) 341-357.
Comment 4: Do not include the chemical formulas of compounds in the title of the manuscript and Keywords.
Response: Thanks a lot for your careful review. According to your advices, we have revised the title of the manuscript and Keywords, rephrased the chemical formulas “CH2O” as “formaldehyde”.
Comment 5: The manuscript section should not begin with a figure or table.
Response: Thanks a lot for your careful review. According to your advices, we have made corresponding improvements about the figure and table.
Comment 6: Figure 4 b2: Why is the interaction of carbon monoxide with phthalocyanine shown in the figure?
Response: Thanks a lot for your careful review. After CH2O gas adsorption, the two H atoms of CH2O gas has been completely dissociated, only the O and C atoms still remain in the gaseous phase. Thus, after CH2O gas molecule adsorption on phthalocyanine, only the O and C atoms remain in the gaseous phase, not the carbon monoxide.
Comment 7: Figure 5, 6 Figure captions should be on the same page as them.
Response: Thanks a lot for your careful review. According to your advices, we have made corresponding improvements about the figure 5 and 6.
Comment 8: The writing needs some revision by native English speaker.
Response: Thanks a lot for this valuable suggestion. According to your advices, we have consulted some native English speakers to revise our manuscript.
We tried our best to improve the manuscript and made the corresponding changes in the manuscript. We appreciate for your warm work earnestly, and hope that the correction will meet with approval. Once again, thank you very much for your comments and suggestions.
Thanks again for your kind consideration
Yours sincerely,
Zhifeng Nie,
Yunnan Key Laboratory of Metal-Organic Molecular Materials and Device, Kunming University, Kunming 650214, China
E-mail: niezf123@163.com

Round 2
Reviewer 3 Report
I consider that the paper has been improved according to Reviewer’s recommendations. I recommend this paper to be accepted for the publication in Journal «Metals».
Best regards, Reviewer